# Non-Parameterized Randomization for Environmental Generalization in Deep Reinforcement Learning

## Abstract

The generalization problem presents a major obstacle to the practical application of reinforcement learning (RL) in real-world scenarios, primarily due to the prohibitively high cost of retraining policies. The environmental generalization, which involves the ability to generalize RL agents to different environments with distinct generative models but the same task semantics, remains an unsolved challenge that directly affects real-world deployment. In this paper, we build a structured mathematical framework to describe environmental generalization and show that the difficulty comes from a non-optimizable gap without learning in all environments. Accordingly, we propose a kind of non-parameterized randomization method to augment the training environments. We theoretically demonstrate that training in these environments will give an approximately optimizable lower bound for this gap. Through empirical evaluation, we demonstrate the effectiveness of our method in zero-shot environmental generalization tasks spanning a wide range of diverse environments. Comparisons with existing advanced methods designed for generalization tasks demonstrate that our method has significant superiority in these challenging tasks. [1]

## 1 Introduction

Reinforcement learning (RL) has emerged as a promising approach for addressing real-world application problems (Mnih et al., 2013; Sutton & Barto, 2018), however, suffers from poor sample efficiency and poor generalization abilities Ghosh et al. (2021); Malik et al. (2021); Huang et al. (2021). This stems from the inherent nature of RL frameworks, where training and testing are tightly integrated. Consequently, RL policies are highly task-specific, and their applicability to analogous tasks is limited. This challenge increases with the growth in task numbers, leading to an exponential explosion in sample requirements and corresponding costs. Thus, improving the generalizing ability of the agent can enhance sample efficiency and make RL more practicable in real-world scenarios.

In practical scenarios, RL agents frequently need to adapt to diverse environmental conditions, necessitating policy adaptations to changes in state space, action space, and transition functions. This requirement, termed "environmental generalization" in RL, remains a complex challenge. Recent works focus on addressing generalization problems, such as Epistemic MDPs (Ghosh et al., 2021), Block-MDPs (Zhang et al., 2020; Han et al., 2021), and the work by Malik et al. (2021), attempt to model generalization problems and formulate corresponding learning algorithms. However, these approaches are limited by the assumption of shared state space across tasks, contradicting the premise of environmental generalization. As solving the environmental generalization problem holds significant potential for enabling more complex real-world applications, our work focuses on this specific challenge within RL policies and aims to make progress in solving it.

The difficulty of achieving environmental generalization within RL has not been adequately analyzed in prior work. In this research, we aim to solve this difficulty by introducing a framework explicitly designed for handling the environmental generalization problem. Our framework involves utilizing a decoupled structurized state space, which allows us to explicitly model the common components and task-agnostic backgrounds. This framework can homogeneously depict both

---

[1]The code is available in the Supplemental Materials.

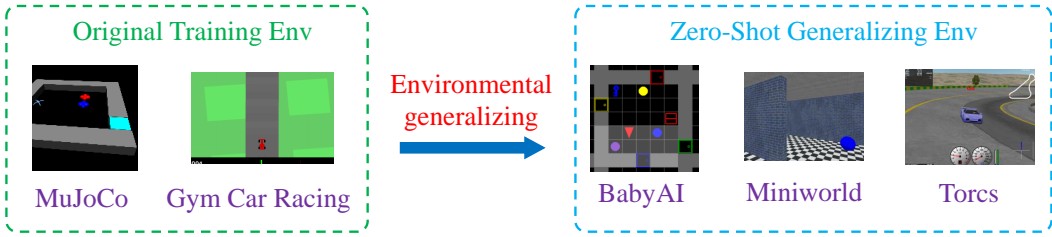

Figure 1: Illustration of Zero-shot environmental generalization tasks. The generalizing tasks are intrinsically different from the training tasks, hence are extremely difficult to solve in zero-shot.

the similarities and differences of tasks across various environments. We observe that successful decision-making in unseen tasks requires the agent to accurately identify the invariant components that represent the task goals, while concurrently ignoring task-agnostic changes in the environment. However, achieving this goal necessitates an exhaustive exploration of all environments, which is impractical. This is because the conventional objective function commonly used in RL methods, i.e., maximizing the return, motivates the agent to overfit the specific dynamics and observations of the environment. It conflicts with the generalization setting, leading to a non-optimizable gap that hinders environmental generalization. We refer to this gap as the ***adaption gap***.

Addressing the non-optimizable gap involves enhancing the agent's adaptive capability without exhaustive environmental traversal. This requires refining the objective function. Existing randomization methods like Automatic Data Augmentation (ADA)(Raileanu et al., 2021) and Domain Randomization (DR) methods (Tobin et al., 2017), which generate multiple training environments to boost the RL agent's adaptability, offer a promising approach. However, methods like DR are reliant on the parameterized dynamics model of the environment for parameter randomization. This reliance restricts the breadth of generalization across diverse environments and introduces additional modeling errors. Thus, we propose a *non-parameterized randomization* (NPR) method. Our approach diverges from previous methods by randomizing task-agnostic components and adding disturbances without the requirement for parameterized models. This divergence implies that our method is not limited to any specific environmental model, thus allowing for adaptation to a broader range of environmental changes. Our theory shows that such intrinsic non-parameterized randomization is equal to introducing an alternative objective function. This objective function serves as an optimizable lower bound for the non-optimizable adaption gap, thereby significantly enhancing the generalization ability towards unseen environments without retraining.

To demonstrate the superiority of our method, we propose challenging environmental generalization tasks by modifying existing complex benchmarks. These tasks are in environments that are intrinsically different, even having different observing views and transition dynamics. To the best of our knowledge, we are the first to achieve generalization tasks with environmental change in zero-shot.

In summary, our contributions are as follows:

1. To the best of our knowledge, our work is the first to introduce a structured framework that uniformly describe the environmental generalization problem. This framework enables the analysis of the inherent challenges in accomplishing such generalization tasks.

2. We propose a novel non-parameterized randomization (NPR) method to tackle environmental generalization. Our theoretical analysis substantiates that this approach can enhance generalization capabilities across unseen environments without necessitating retraining.

3. We have designed challenging experiments for environmental generalization across a broad range of prevailing environments. The empirical results, compared with advanced baselines in intricate zero-shot tasks, demonstrate the superiority of our method.

## 2  RELATED WORK

**Background of Generalization Works in RL**. Building generalizable policies that can be reused in new tasks is a long-standing challenge. There are many RL and HRL works that focus on generalization tasks. Theoretically, there are works like Wang et al. (2019) describing a common gap of different tasks in the RL domain, Ghosh et al. (2021) giving the tractable generalizing conditions in meta-RL and modeling the generalization tasks as POMDPs (Ghosh et al., 2021). Methodologically,

there are some methods utilizing injected noise in the observation space (Raileanu et al., 2021) or in the dynamics model of the environment (Tobin et al., 2017), and introducing additional input of shared languages or symbols (Jiang et al., 2019; Vaezipoor et al., 2021) to improve the generalization ability of the agent. Among them, the most related works are as follows:

**Context-Conditioned MDPs**. Some RL works model the generalization tasks as utilizing learned shared knowledge to deal with new similar tasks. As a result, they leverage shared prior as additional input like language (Chen et al., 2020), or build context (Levy & Mansour, 2023) or meta-learning process (Kirsch et al., 2019b;a) to make the policies more adaptable. However, all the existing works have an assumption that these tasks should be similar in the environmental aspect. That means these works are limited. In this paper, our work focuses on generalization tasks with environmental changes, which attempts to improve the existing assumption and build more generalizable policies.

**Randomization as Augmentation**. Some RL works utilize injected noise in the learning process to improve the adaptive capability of the agent (Gur et al., 2021; Fan et al., 2021), such as observation augmentation (Raileanu et al., 2021) and domain randomization (DR) (Tobin et al., 2017). The former methods inject noise in the observation space after sampling, which we call external randomization, and can hardly cover the change of environment structures. The latter methods, making intrinsic noise in environments including visual and dynamic randomization, usually require a parameterized model to describe the change in the target environments. It cannot solve the OOD change that cannot be described by the parameterized models. Different from previous methods, our work intrinsically randomizes the environment to build task-level augmentations and does not require a specific parameterized model.

**Learning Invariable Representation**. Some works try to learn shared representation and build reusable policies, like learning causal invariant representation in Block MDPs (Zhang et al., 2020; Han et al., 2021), or learning representations as reusable subgoals in HRL works (Liu et al., 2020). This works based on the assumption that there exists some shared states or a whole shared state space. The differences in different tasks are just caused by different views of partial observation. Thus, if the shared states are extracted, they can build reusable policies. However, this assumption does not hold in generalization tasks that possess significant change caused by the intrinsic difference of the environments. That means the shared parts in different environments cannot directly be aligned for executing the learned decision. In this work, we aim to extend the setting of previous works and focus on more widely generalizing tasks, hence loosening the existing assumption.

# 3 MODEL AND ANALYSIS FOR ENVIRONMENTAL GENERALIZATION

## 3.1 STRUCTURIZED MODEL FOR ENVIRONMENTAL GENERALIZATION

In this section, we will introduce the setting of environmental generalization problems. To model such generalization tasks, we propose a structurized model to uniformly describe the state space in different environments. Thus we can discuss the tasks in different environments uniformly. Here we mainly focus on the change of state space and assume the action spaces are the same. All the proof can be seen in Appendix A.

**Preliminary**. We formulate the task in this paper as a goal-conditioned Markov decision process (MDP) in multiple environments, defined as a tuple $\mathcal{M}^e(I) = < \mathcal{S}^e, \mathcal{A}, P^e, R^e, \gamma >, e \in \mathcal{E}, I \in \mathcal{I}$. Here $\mathcal{E}$ is the set of existing environments. $\mathcal{S}^e$ is the state space of environment $e$ and $\mathcal{A}$ is the action space. $\mathcal{I}$ is the shared representation space, where the representation $I$ stays invariable and represents the common points of the same task in different environments. $P^e$ is the transition probabilities, $R^e$ is the reward function. The goal of the agent is to learn a goal-conditioned policy $\pi(a_t|s_t^e, \hat{I})$ to maximize the cumulative return in any tasks of every environment, i.e., $\max_\pi \mathbb{E}_{I \in \mathcal{I}, e \in \mathcal{E}}[\sum_{t \geq 0} \gamma^t R^e(s_t^e, a_t|I)]$. Here $\hat{I}$ is the given representation to distinguish the task. When generalizing, the environments $e$ are not all available in the training process.

**Existed Modeling Challenge**. In environmental generalization tasks, there is a challenge that the existing problem modeling methods do not help deal with such tasks. That is because the states in different environments are quite different, leading to significant discrepancies in the input to the agent. Thus, how to measure the similarity and difference of a task in different environments becomes a challenge. In this paper, we propose a structurized model to give a decoupled expression

of the common parts and differences as follows. By this model, we can accordingly explain why this problem is so difficult and how we are inspired to mitigate it.

**Structurized Model**. Consider that in MDPs in different environments, the observations are quite different. Here we will focus on the tasks that have intrinsic common points, which have different forms. We first give a new modeling form of observation to represent the similarities and differences of different tasks by structuring the state space. The definition is as follows:

**Definition 3.1.** *(Structurized State Space) Consider* $\forall e \in \mathcal{E}$*, the structurized state can be written as:*

$$s_t^e = \psi_t(I) \oplus \xi_t^e \tag{1}$$

*where $\psi_t$ is a reversible function depending on the current step in the environment, $I \in \mathcal{I}$ is the shared representation among all the environments of in any task, $\xi_t^e$ is a task-agnostic background which only depends on the environment.*

This definition is utilized to describe the state in different complex scenes. Consider that in real-world problems, the states are usually structured and can be composed of task-dependent objects or goals and task-independent backgrounds. This formulation can represent almost all kinds of state spaces, where previous works can be seen as special cases of ours with fixed backgrounds in specific tasks. Meanwhile, the function $\psi_t$ means that the invariant of the task is not always observable in all the steps, which can also describe the situations of partial observation tasks.

**Difference with Previous Models**. Some works also utilize a structured state space like Block-MDPs (Zhang et al., 2020). In previous works, the observation space is a part of the shared state space caused by partial observation. That means their observation can naturally be aligned with different tasks. But in more complex generalization tasks, especially in real-world applications, the common parts of tasks are usually embedded in the environment and not always observable. So the environmental generalization tasks have diverse state space and cannot be easily aligned. As a result, in environmental generalization tasks, there is extra difficulty in extracting the common parts and aligning them. Thus, different from existing works, our model describes states in different environments with a decoupled model to describe the similarity parts (the $\psi_t(I)$) and changing background (the $\xi_t^e$) of state space in Def 3.1, where the similarity parts (the $I$) are embellished by the environment (the $\psi_t(\cdot)$) but invariable.

## 3.2 ANALYSIS OF CHALLENGES IN ENVIRONMENTAL GENERALIZATION

In this section, we will analyze why environmental generalization problems are so difficult. According to our mathematical model, there is a non-optimizable gap between different environments. All the proof can be seen in Appendix A.

As said above, learning to extract invariable representation to build a generalization policy is difficult. In this section, we will give an analysis of why it is difficult and how to deal with it.

**Error Analysis towards Generalization**. To describe the difficulty of generalizing to different environment, without loss of generality, firstly we consider the error of two value functions of the same task in different environments, i.e., $|V_t^{e_1}(s_t^{e_1}|I_1) - V_t^{e_2}(s_t^{e_2}|I_1)|$ for any $e_1, e_2 \in \mathcal{E}$, where $V_t^{e_1}(s_t^{e_1}|I_1) = \sum_{s_{t+1}^{e_1}} \sum_{a_t} P^{e_1}(s_{t+1}^{e_1}|s_t^{e_1}, a_t)\pi(a_t|s_t^{e_1}, \hat{I}_1)(R(s_t^{e_1}, a_t|I_1) + \gamma V_{t+1}^{e_1}(s_{t+1}^{e_1}|I_1))$.

In the components of the value function, there are naturally two important parts, the transition $P^{e_1}(s_{t+1}^{e_1}|s_t^{e_1}, a_t)$ depend on the environment and the policy $\pi(a_t|s_t^{e_1}, \hat{I}_1)$.

Considering humankind's decisions in real-world tasks, we will always make similar decisions in similar tasks, ignoring the task-agnostic background. Inspired by this phenomenon, we consider it reasonable to measure the similarity of policies in the invariant representation space:

**Assumption 3.2.** *(Invariant Metric) For two well-learned policies from two environments, the difference can be measured in the representation space as:*

$$|\pi^{e_1}(a_t|s_t^{e_1}, I_1) - \pi^{e_2}(a_t|s_t^{e_2}, I_2)| \le L_\psi \|I_1 - I_2\| \tag{2}$$

By this metric, there is a natural corollary that if the tasks in different environment are the same, the policy should also be same, i.e., $\pi^{e_1}(a_t|s_t^{e_1}, I_1) = \pi^{e_2}(a_t|s_t^{e_2}, I_2)$ when $I_1 = I_2$. It satisfies the common sense said above.

Different from existing works that make policy metrics in original state space like (Wang et al., 2019), our metric can cover more situations with more complex states, where the distances in the original state space are usually inaccessible or meaningless. For instance, in a high-dimension state space that represents the parameters of joints of a robot like MuJoCo (Todorov et al., 2012), the highly non-linearity makes the distance in the original state space helpless to measure the difference of different policies.

By assumption 3.2, we can give a generalization error bound which can be used in any generalizing scenes with environmental changes:

**Proposition 3.3.** *(Environmental Generalization Error) With discounted factor $\gamma$ and bounded reward function $\max_{s,a,e,I} R(s^e, a^e|I) = R_{max}$, Lipschitz constant $L_\psi$, for any environments $e_1, e_2 \in \mathcal{E}$, there is:*

$$\max_{e_1,e_2\in\mathcal{E}} |V_t^{e_1}(s_t^{e_1}|\hat{I}_1) - V_t^{e_2}(s_t^{e_2}|\hat{I}_2)| \leq \frac{R_{max}}{(1-\gamma)^2}.$$

$$\left[\underbrace{L_\psi |A| \cdot \|\hat{I}_1 - \hat{I}_2\|}_{invariant\ learning\ error} + \underbrace{\max_{e_1,e_2,e} |S^e|^2 \left| \frac{P(s_{t+1}^{e_1}|s_t^{e_1}, a_t)}{|S^{e_2}|} - \frac{P(s_{t+1}^{e_2}|s_t^{e_2}, a_t)}{|S^{e_1}|} \right|}_{adaption\ gap}\right] + \frac{R_{max}}{1-\gamma} \tag{3}$$

*where $|\cdot|$ is the cardinality, $\hat{I}_1$ and $\hat{I}_2$ is the learned representation from the same representation $I$ in different environments.*

This theorem shows that there are two independent parts when generalizing from one environment to another. However, they are quite different, because one of them is optimizable but the other is not. Shown as the following proposition, the invariant learning can be solved by providing an instruction $\hat{I}$ and making it consistent with the reward that represents the goal of the task. By this, training the policy in tasks with an invariable instruction depending on the invariant representation will implicitly build a mapping policy from the given instruction to the states that represent the task. After that, the agent can identify the task by the given instruction, instead of requiring the real representations.

**Proposition 3.4.** *(Implicit Invariant Learning) With sparse reward 1 of the final state representing completing the task, maximizing the expected training return of $\pi(\hat{I})$, is equal to maximizing the occurrence of the invariable shared part of the same task in a different environment.*

$$\max_{\pi(\hat{I})} \mathbb{E}_{e\in\mathcal{E}, \tau^e \sim \pi^e} \left[\sum_{t \geq 0} \gamma^t R^e(s_t^e, a_t|I)\right] = \max_{\pi(\hat{I})} P^\pi(I|\hat{I}) \tag{4}$$

**Non-optimizable Gap**. Attention that the adaption gap can not be directly optimized, because it only depends on the distribution of the background of the environments which are unseen when generalizing. Even building a transition predicting model by model-base RL methods is not enough, due to the uncertainty of the unseen generalization environment with out-of-distribution data.

Our analysis and framework highlight the extreme difficulty of achieving environmental generalization in complex RL tasks. Although existing methods have successfully obtained generalizing capability in some specific tasks, they cannot deal with this problem. A more effective method is necessary for learning policies that perform well in environmental generalization tasks.

# 4 NON-PARAMETERIZED RANDOMIZATION FOR ENVIRONMENTAL GENERALIZATION

## 4.1 THE NON-PARAMETERIZED RANDOMIZATION (NPR) METHOD

**Feasibility of NPR**. With the analysis above, we can see that the key to solving environmental generalization tasks is to deal with the adaption gap that is unable to be directly optimized. In this paper, we propose a novel idea that introduces random noise into the task-agnostic components in the training environments to approximate the change in the environment.

Specifically, the state $s_t^e = \psi_t(I) \oplus \xi_t^e$ are not available for all the environment $e$ in generalization tasks, meaning that $\xi_t^e$ cannot be exhaustively explored. We will replace the task-agnostic part $\xi_t^e$

Figure 2: Comparison of our method and existing augmentation methods.

with the randomized background $\hat{\xi}_t$. Here $\hat{\xi}_t$ is not parameterized, hence is not limited to the parameterized model of the environment. Training with tasks in randomized environments is utilized to motivate the agent to overcome the task-agnostic disturbances focus on the invariable task representation and make similar decisions. It can be seen as a kind of task-level data augmentation that generates more tasks in different approximated environments by randomization. To prove the feasibility, we give the theorem as follows. The proof can be seen in Appendix A.

**Theorem 4.1.** *(Approximating Feasibility) For a set of backgrounds with injected noise denoted as $\hat{\xi}_t \in \Xi$ and corresponding generated state denoted as $\hat{s}_t$, with bounded reward functions $\hat{R}_{max} = \max_{e,\hat{\xi}_t,a_t,I}\{R(s_t^e, a_t|I), R(\hat{s}_t, a_t|I)\}$, there is:*

$$\mathbb{E}_{e \in \mathcal{E}, \tau^e \sim \pi^e}[\sum_{t \geq 0} \gamma^t R(s_t^e, a_t|I)] \geq \mathbb{E}_{\hat{\xi} \in \Xi, \hat{\tau} \sim \hat{\pi}}[\sum_{t \geq 0} \gamma^t R(\hat{s}_t, a_t|I)] - \alpha \tag{5}$$

*where $\alpha = \frac{1}{1-\gamma}(\hat{R}_{max}\sqrt{2D_{KL}(\rho(e)||\rho(\hat{\xi}_t))} + \delta_{max})$ is a constant depending on the similarity of the augmented environments and the unseen environments. Here $\rho(e)$ and $\rho(\hat{\xi})$ represent the distributions of the unseen environments and the randomized environments. $\delta_{max} = \max_{e,\hat{\xi}_t}|R(s_t^e, a_t|I) - R(\hat{s}_t, a_t|I)|$*

This theorem gives an exciting result that, if the injected noise conforms to the change of the environments, $\alpha$ will be a little constant and can be ignored, meaning that learning in randomized environments can be seen as approximately maximizing the lower bound of the original return. It shows that training the agent in the randomized environment will also improve the generalization ability, even if the generalizing environments are unseen and the trained environments are different from the generalizing ones. This theorem indicates that we can leverage injected noise as a substitute for training in real environments and save the cost of sampling.

**Remark 4.2.** *In the proving process, we found that if utilizing a parameterized model to generate the environments, the lower bound of 4.1 will add another term caused by the discrepancy of the models of the generalizing environments and the original environments. Meanwhile, $\delta_{max}$ depends on a one-step return due to our problem setting. If utilizing a parameterized model different from the generalizing environment, $\delta_{max}$ will be larger in another form. This fact supports our claim that existing randomization methods relying on the parameterized models will perform poorly in dealing with environmental change.*

## 4.2 Implementation of NPR in RL Environments

**Implementation of NPR method**. We design an intrinsic model-free randomization method to build training tasks. Specifically, we randomize the existing components in the environments (intrinsic randomization), instead of injecting noise in the color of the observation like ADA or in the parameters of the environment model like DR-like methods (external noises) (See in Figure 2). External augmentations usually cannot represent the change in environments, and DR-like methods are always limited to the parameterized model. To make our idea more general, we propose to improve DR methods. That is, to randomize the non-parameterized task-agnostic parts of the training environments, like randomizing the structure of the environment, randomizing the background by adding additional task-agnostic disturbance and randomizing the spatial relationship of all the existing objects.

For instance, in a kitchen, if the robot should find an apple, we can randomize all the task-agnostic elements in the kitchen like the microwave oven, the refrigerator, the structure of the room, the position of the apple, and add some unrelated objects as a disturbance. With the various disturbances and the apple staying invariable, the agent is forced to learn to overcome the environmental change

and obtain the apple. Then it will also obtain the apple in an unseen environment by seeing the background as noise. Similarly, training a car agent to race on roads with changeable shapes will force it to learn to keep on the road, which will significantly improve the adaptability to unseen roads with different but similar dynamics.

It can be seen that our method aims to randomize the non-parameterized elements in the environments. It will encourage the agent not to limit to the parameter spaces. The advantage is that in any environment it is effective because it does not require the parameterized model of the environment. The disadvantage is that it needs expert priors. But we consider that if the policy can be reused in many unseen new tasks, the disadvantage is acceptable in real-world deployment.

**Soft Randomizing and Parallel Learning Algorithm**. As we know, training the agent in dynamic environments usually causes learning instability. Because compared with fixed environments, the unacceptable large variance in the dynamic learning process will disturb the gradient convergence direction. Therefore, for stable learning, the random noise should not be arbitrary. Thus, to make the learning process stable, we utilize soft randomizing with a continuous and slow episodic change to reduce the variance of learning, which accords with the analysis above. We also utilize parallel online learning algorithms to reduce learning instability because there are some works that show the potential of parallel algorithms to adapt to dynamic environments (Hou et al., 2022). We use actor-critic-like algorithms with our randomization method for tasks in discrete environments and PPO algorithms for tasks in continuous environments. Details of the algorithm can be seen in Appendix C.

## 5 EXPERIMENTS

### 5.1 EXPERIMENTS SETTINGS

**Generalization Experiments**. As there are no works that have solved environmental generalization tasks, we utilize several prevailing environments to build generalization experiments across different environments, including MuJoCo (Todorov et al., 2012), gym (Towers et al., 2023), Torcs (Loiacono et al., 2013), and BabyAI (Chevalier-Boisvert et al., 2018). In these tasks, the training tasks and testing tasks are in different environments and do not allow retraining, to show the zero-shot generalization capabilities. The training tasks are evaluated by reward curves and the generalizing tasks are evaluated by zero-shot success rates and zero-shot rewards. The details of the environments and randomization can be seen in Appendix B.

1. The agent is trained in the MuJoCo environment and then generalizes to the new mazes of BabyAI in zero-shot. The goals of the tasks are all to navigate or to find an object. The differences are disparate observation space and different motion dynamics. The action space is discrete and executed by the simulator. The randomized components are the structure of the room, relative position, and unrelated objects. Such environmental generalizing tasks have not been solved by existing works.

2. Training in a simple 2D car racing game in the gymnasium, the agent should generalize to a new complex 3D car racing game close to the real-world scene in Torcs in zero-shot. The differences are disparate observation space and different motion dynamics. The agent should keep on the roads with different shapes and go forward to obtain more rewards. The randomized components are the shape of the track, the zoomed viewpoints, and the background. Such environmental generalizing tasks have not been solved by existing works.

**Baselines**. As the generalization tasks in this paper have significant changes in the environment, they can be hardly reflected in the vectorized observation space. As a result, we choose the most advanced pixel-based RL methods as baselines for a fair comparison.

1. Classical RL method like PPO (Schulman et al., 2017) and advanced RL method like Droq (Hiraoka et al., 2021) with pixel observation by CNN. Comparing these universal advanced methods will show the superiority of our method in generalization tasks.

2. Pixel-based RL SOTA augmentations method in observation for generalization like DrAC (Raileanu et al., 2021) and intrinsic randomization like DR (Tobin et al., 2017). These

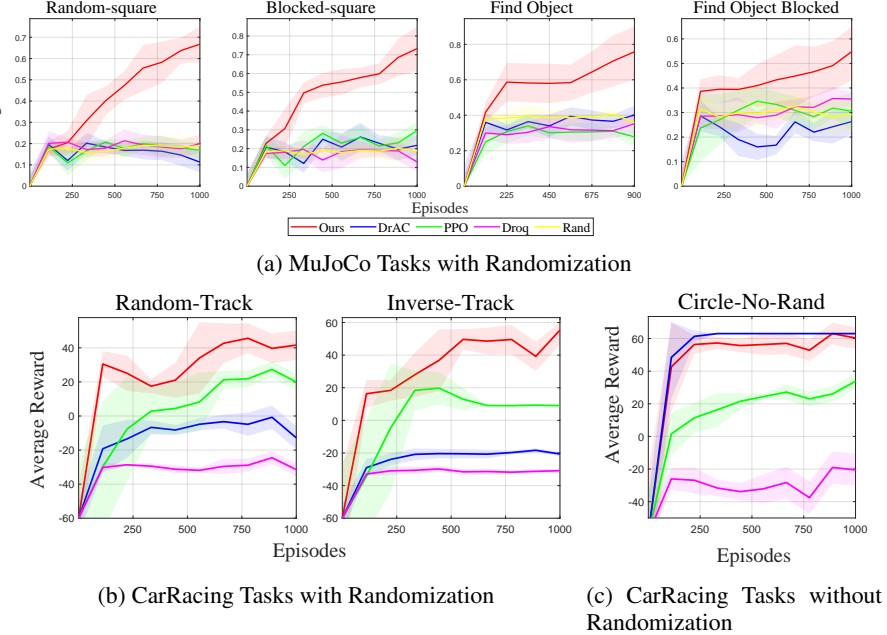

(a) MuJoCo Tasks with Randomization

(b) CarRacing Tasks with Randomization

(c) CarRacing Tasks without Randomization

Figure 3: Comparative Experiment Results. The results are averaged by three random seeds with $\pm$ a standard deviation. The legend bar in (a) is shared.

methods also focus on generalization tasks. Comparing this method will directly show the differences between augmenting methods and show the superiority of our method.

## 5.2 RESULTS

**Stable Learning**. Firstly, we will show the reward curves in the learning process to show the learning stability. We emphasize again that training stably in randomized environments with noise is not easy. The results of learning in randomized tasks are shown in Figure 3. Especially, compared with learning in tasks with randomization (Figure 3b) and the task without randomization (Figure 3c), we can see that the baselines perform well in common tasks but perform poor in tasks with randomization, sometimes even worse than a random policy (The yellow curve in 3a). That means they cannot adapt to the randomness of environmental changes well. On the contrary, our method can learn stably in randomized tasks. It shows the superiority of our method to build adaptable policy in a stable learning process.

Table 1: Generalization Task for Navigation and Object Interaction (Zero-shot Success Rate %).

| Method | Trained Randomized Tasks | | Trained Envs Unseen Tasks Generalization | | Unseen Envs Unseen Tasks Environmental Generalization | |
|---|---|---|---|---|---|---|
| | Maze | FindObj | Maze-g1 | Interaction1 | Maze-g2 | Interaction2 |
| **Ours** | **70.8** $\pm$ 4.2 | **74.8** $\pm$ 6.3 | **38.0** $\pm$ 8.3 | **68.4** $\pm$ 5.1 | **30.4** $\pm$ 4.2 | **34.2** $\pm$ 7.1 |
| DrAC | 12.8 $\pm$ 6.4 | 35.4 $\pm$ 4.6 | 0.8 $\pm$ 1.3 | 15.6 $\pm$ 12.1 | 3.4 $\pm$ 2.1 | 9.3 $\pm$ 2.8 |
| PPO | 18.6 $\pm$ 4.2 | 31.8 $\pm$ 4.6 | 1.2 $\pm$ 0.8 | 28.2 $\pm$ 4.5 | 7.6 $\pm$ 2.9 | 23.4 $\pm$ 4.0 |
| Droq | 18.8 $\pm$ 3.1 | 37.4 $\pm$ 3.7 | 0.6 $\pm$ 0.9 | 36.8 $\pm$ 3.6 | 12.8 $\pm$ 4.2 | 0.2 $\pm$ 0.5 |
| No-Rand | — | — | 0.0 $\pm$ 0.0 | 2.6 $\pm$ 1.8 | 0.2 $\pm$ 0.5 | 2.8 $\pm$ 3.0 |

**Generalization in Zero-Shot**. To sufficiently show the generalization capability of these methods, we divide the generalization task into two parts, i.e., in-domain generalization for unseen tasks in the same environment, and out-of-distribution generalization for tasks in different unseen environments. We utilize the agent trained in 'Random-Square','  Find-Obj', and 'Random-Track' to generalize to unseen tasks in different environments ***without retraining***. Results in Table 1 are tested in 500 episodes, and results in Table 2 are tested with 5 seeds. The details of these tasks can be seen in the Appendix B.

The results can be seen in Table 1 and Table 2. All the generalization tasks are significantly different from the training tasks. We can see that, in these generalization tasks, even in unseen new environments, our agent can still complete the tasks with the highest success rates and rewards. However, the baselines cannot adapt to the change in the environment and perform poorly. Including the DR method with visual randomization (Table 2), it performs poorly in environmental generalizations due to the dependence on learned environment models. It shows the superiority of our method in building environmental generalizable policies.

Table 2: Generalization Task for Car Racing (Zero-Shot Average Reward).

| Method/Racing | Generalization in Unseen Environments | | |
| --- | --- | --- | --- |
| | CG-Track2 | Street1 | Alpine1 |
| Ours | $\mathbf{597.7} \pm 31.4$ | $\mathbf{1108.9} \pm 653.0$ | $\mathbf{1651.1} \pm 1037.8$ |
| No-Rand | $233.9 \pm 170.7$ | $353.9 \pm 409.8$ | $834.6 \pm 664.3$ |
| DrAC | $281.5 \pm 134.9$ | $35.25 \pm 0.52$ | $162.5 \pm 11.1$ |
| PPO | $254.46 \pm 100.5$ | $217.5 \pm 324.6$ | $71.8 \pm 14.2$ |
| PPO + DR | $455.0 \pm 91.1$ | $601.3 \pm 245.0$ | $1001.0 \pm 612.7$ |

## 5.3 ABLATION STUDY

**Poor Generalization without Randomization**. To show the effects of randomization, we add a comparison in generalization experiments with direct learning in original training tasks without randomization of our agent (the baseline 'No-Rand'). These tasks are fixed without randomness like the common RL setting. As the training tasks are different from the randomized tasks, we only show the results in generalization tasks. It can be seen that the agent trained without randomization completely cannot generalize to the new tasks, both in the same environment and in different environments.

## 5.4 CHALLENGING ENVIRONMENTAL GENERALIZATION VERIFICATION

We argue that our method has the potential to be utilized in real-world applications due to its strong generalization ability. To show it, we design an extremely challenging generalization task, i.e., training in the 2D room of 3rd personal view (MuJoCo) and generalize to the 3D room of the 1st personal view (MiniWorld) in zero-shot.

Table 3: Challenging Generalization Task (Zero-shot Success Rate %)

| Method | Ours | No-Rand |
| --- | --- | --- |
| Training-2D-maze | $71.9 \pm 4.5$ | $100.0 \pm 0.0$ |
| Generalizing-3D-maze | $66.6 \pm 5.3$ | $0.6 \pm 0.6$ |

We design this task as an identification task to let the agent make a one-step decision to find the correct object. It can be seen in Table 3 that our agent has the probability to identify the goal correctly. That means our method can be used to help build a general initial policy to make a high-level decision without retraining.

## 6 CONCLUSION

In this paper, we propose a novel framework that tries to describe and solve the generalization RL tasks that have intrinsic environmental change. To the best of our knowledge, we are the first to discuss and attempt to deal with this problem. We believe that, in the future, our ideas will be helpful in building a general RL large model for real-world application as a task-level augmentation method, just like LLM in the NLP domain.

**Limitations and Future Work**. This work focuses on environmental generalization, which is mainly shown in the observation space. In real-world applications, there are many tasks that require significant change in the action space, which will be our future work. Besides, another direction is to leverage more complex semantic representations like nature language from LLM to achieve higher-level generalizations, including long-horizon strategy transferring. This paper provides a scalable port to combine with more modules.

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

# A  THEORETICAL ANALYSIS AND PROOF

## A.1  PROOF OF PROPOSITION 3.3

*Proof.* We recall the setting in that we focus on environmental generalization with structurized goal-conditioned MDPs.

Firstly, we consider the error between two value functions learned in two arbitrarily different environments $e_1$ and $e_2$ with the same task representation $I$. That is:

$$|V^{e_1}(s_t^{e_1}|I) - V^{e_2}(s_t^{e_2}|I)| \tag{6}$$

Consider that, the real representations of the task may not always be obtained, the RL agent will frequently receive given instructions $\hat{I}_1$ and $\hat{I}_2$ that represent task $I$ in different environments. Thus, the value function should be the corresponding policy should be $V^{e_1}(s_t^{e_1}|\hat{I}_1)$ and $V^{e_2}(s_t^{e_2}|\hat{I}_2)$, and the corresponding policies are $\pi^{e1}(a_t|s_t^{e1}, \hat{I}_1)$ and $\pi^{e2}(a_t|s_t^{e2}, \hat{I}_2)$. Then we expend the value function:

$$
\begin{aligned}
&|V^{e_1}(s_t^{e_1}|\hat{I}_1) - V^{e_2}(s_t^{e_2}|\hat{I}_2)| \\
=&|\sum_{s_{t+1}^{e_1}} P(s_{t+1}^{e_1}|s_t^{e_1}, a_t) \sum_{a_t} \pi^{e_1}(a_t|s_t^{e_1}, \hat{I}_1) \left[ R^{e_1}(s_t^{e_1}, a_t|I) + \gamma V^{e_1}(s_{t+1}^{e_1}|\hat{I}_1) \right] \\
&- \sum_{s_{t+1}^{e_2}} P(s_{t+1}^{e_2}|s_t^{e_2}, a_t) \sum_{a_t} \pi^{e_2}(a_t|s_t^{e_2}, \hat{I}_2) \left[ R^{e_2}(s_t^{e_2}, a_t|I) + \gamma V^{e_2}(s_{t+1}^{e_2}|\hat{I}_2) \right] |
\end{aligned}
\tag{7}
$$

Here we temporally denote $[R^{e_1}(s_t^{e_1}, a_t|I) + \gamma V^{e_1}(s_{t+1}^{e_1}|\hat{I}_1)]$ as $G^{e_1}(\hat{I}_1)$. Notice that the state spaces in different environments may be different, here we consider the more complex case that $|S^{e_1}| \neq |S^{e_2}|$. Then Equation 7 cannot be directly dealt with. We treat them by aligning the distribution as follows:

$$
\begin{aligned}
&|V^{e_1}(s_t^{e_1}|\hat{I}_1) - V^{e_2}(s_t^{e_2}|\hat{I}_2)| \\
=&|\sum_{s_{t+1}^{e_1}} P(s_{t+1}^{e_1}|s_t^{e_1}, a_t) \sum_{a_t} \pi^{e_1}(a_t|s_t^{e_1}, \hat{I}_1) G^{e_1}(\hat{I}_1) \\
&- \sum_{s_{t+1}^{e_2}} P(s_{t+1}^{e_2}|s_t^{e_2}, a_t) \sum_{a_t} \pi^{e_2}(a_t|s_t^{e_2}, \hat{I}_2) G^{e_2}(\hat{I}_2)| \\
=&|\frac{\sum_{S^{e_2}}}{|S^{e_2}|} \sum_{s_{t+1}^{e_1}} P(s_{t+1}^{e_1}|s_t^{e_1}, a_t) \sum_{a_t} \pi^{e_1}(a_t|s_t^{e_1}, \hat{I}_1) G^{e_1}(\hat{I}_1) \\
&- \frac{\sum_{S^{e_1}}}{|S^{e_1}|} \sum_{s_{t+1}^{e_2}} P(s_{t+1}^{e_2}|s_t^{e_2}, a_t) \sum_{a_t} \pi^{e_2}(a_t|s_t^{e_2}, \hat{I}_2) G^{e_2}(\hat{I}_2)|
\end{aligned}
\tag{8}
$$

Then there is:

$$
\begin{aligned}
(8) =&|\sum_{S^{e_2}} \sum_{S^{e_1}} \sum_{a_t} \frac{1}{|S^{e_2}|} P(s_{t+1}^{e_1}|s_t^{e_1}, a_t) \pi^{e_1}(a_t|s_t^{e_1}, \hat{I}_1) G^{e_1}(\hat{I}_1) \\
&- \frac{1}{|S^{e_1}|} P(s_{t+1}^{e_2}|s_t^{e_2}, a_t) \pi^{e_2}(a_t|s_t^{e_2}, \hat{I}_2) G^{e_2}(\hat{I}_2)|
\end{aligned}
\tag{9}
$$

By introducing a virtual combined term $\frac{1}{|S^{e_1}|} P(s_{t+1}^{e_2}|s_t^{e_2}, a_t) \pi^{e_2}(a_t|s_t^{e_2}, \hat{I}_2) G^{e_1}(\hat{I}_1)$ in to Equation 9, there is:

$$(8) \leq \text{ⓐ} + \text{ⓑ} \tag{10}$$

where

$$
\begin{aligned}
\text{ⓐ} =& |\sum_{S^{e_2}}\sum_{S^{e_1}}\sum_{a_t} \frac{1}{|S^{e_2}|} P(s_{t+1}^{e_1}|s_t^{e_1},a_t)\pi^{e_1}(a_t|s_t^{e_1},\hat{I}_1)G^{e_1}(\hat{I}_1) \\
& - \frac{1}{|S^{e_1}|} P(s_{t+1}^{e_2}|s_t^{e_2},a_t)\pi^{e_2}(a_t|s_t^{e_2},\hat{I}_2)G^{e_1}(\hat{I}_1)|
\end{aligned}
\tag{11}
$$

$$
\begin{aligned}
\text{ⓑ} =& |\sum_{S^{e_1}}\sum_{S^{e_1}}\sum_{a_t} \frac{1}{|S^{e_1}|} P(s_{t+1}^{e_2}|s_t^{e_2},a_t)\pi^{e_2}(a_t|s_t^{e_2},\hat{I}_1)G^{e_1}(\hat{I}_1) \\
& - \frac{1}{|S^{e_1}|} P(s_{t+1}^{e_2}|s_t^{e_2},a_t)\pi^{e_2}(a_t|s_t^{e_2},\hat{I}_2)G^{e_2}(\hat{I}_2)|
\end{aligned}
\tag{12}
$$

Here we deal with them separately. Consider that all the rewards $R > 0$ holds, then $G > 0$. There is:

$$
\begin{aligned}
\text{ⓐ} \leq & \sum_{S^{e_2}}\sum_{S^{e_1}}\sum_{a_t} G^{e_1}(\hat{I}_1) \cdot \\
& \underbrace{\left|\frac{1}{|S^{e_2}|} P(s_{t+1}^{e_1}|s_t^{e_1},a_t)\pi^{e_1}(a_t|s_t^{e_1},\hat{I}_1) - \frac{1}{|S^{e_1}|} P(s_{t+1}^{e_2}|s_t^{e_2},a_t)\pi^{e_2}(a_t|s_t^{e_2},\hat{I}_2)\right|}_{\text{ⓒ}}
\end{aligned}
\tag{13}
$$

Similarly consider ⓒ:

$$
\begin{aligned}
\text{ⓒ} \leq & \left|\frac{1}{|S^{e_2}|} P(s_{t+1}^{e_1}|s_t^{e_1},a_t)\pi^{e_1}(a_t|s_t^{e_1},\hat{I}_1) - \frac{1}{|S^{e_2}|} P(s_{t+1}^{e_1}|s_t^{e_1},a_t)\pi^{e_2}(a_t|s_t^{e_2},\hat{I}_2)\right| \\
& + \left|\frac{1}{|S^{e_2}|} P(s_{t+1}^{e_1}|s_t^{e_1},a_t)\pi^{e_2}(a_t|s_t^{e_2},\hat{I}_2) - \frac{1}{|S^{e_1}|} P(s_{t+1}^{e_2}|s_t^{e_2},a_t)\pi^{e_2}(a_t|s_t^{e_2},\hat{I}_2)\right| \\
= & \frac{1}{|S^{e_2}|} P(s_{t+1}^{e_1}|s_t^{e_1},a_t)\left|\pi^{e_1}(a_t|s_t^{e_1},\hat{I}_1) - \pi^{e_2}(a_t|s_t^{e_2},\hat{I}_2)\right| \\
& + \pi^{e_2}(a_t|s_t^{e_2},\hat{I}_2)\left|\frac{1}{|S^{e_2}|} P(s_{t+1}^{e_1}|s_t^{e_1},a_t) - \frac{1}{|S^{e_1}|} P(s_{t+1}^{e_2}|s_t^{e_2},a_t)\right|
\end{aligned}
\tag{14}
$$

With Equation 13 and Equation 14, since that

$$|G^{e_1}(\hat{I}_1)| = |R^{e_1}(s_t^{e_1},a_t|I) + \gamma V^{e_1}(s_{t+1}^{e_1}|\hat{I}_1)| \leq \frac{R_{max}}{1-\gamma} \tag{15}$$

where $R_{max} = \max_{s,a,e,I} R(s^e,a^e|I)$, there is:

$$
\begin{aligned}
\text{ⓐ} \leq & \frac{R_{max}}{1-\gamma} \cdot \\
& \left[\sum_{a_t}\left|\pi^{e_1}(a_t|s_t^{e_1},\hat{I}_1) - \pi^{e_2}(a_t|s_t^{e_2},\hat{I}_2)\right| + \sum_{S^{e_2}}\sum_{S^{e_1}}\left|\frac{1}{|S^{e_2}|} P(s_{t+1}^{e_1}|s_t^{e_1},a_t) - \frac{1}{|S^{e_1}|} P(s_{t+1}^{e_2}|s_t^{e_2},a_t)\right|\right]
\end{aligned}
\tag{16}
$$

As for ⓑ, there is:

$$
\begin{aligned}
ⓑ &\le \max_{e_1,e_2} |G^{e_1}(\hat{I}_1) - G^{e_2}(\hat{I}_2)| \\
&\le R_{max} + \gamma \max_{e_1,e_2,t} |V^{e_1}(s_{t+1}^{e_1}|\hat{I}_1) - V^{e_2}(s_{t+1}^{e_2}|\hat{I}_2)|
\end{aligned}
\tag{17}
$$

Then we rewrite Equation 8:

$$
\begin{aligned}
|V^{e_1}(s_t^{e_1}|\hat{I}_1) - V^{e_2}(s_t^{e_2}|\hat{I}_2)| \le \frac{R_{max}}{1-\gamma} \cdot \\
\left[ \sum_{a_t} \left| \pi^{e_1}(a_t|s_t^{e_1}, \hat{I}_1) - \pi^{e_2}(a_t|s_t^{e_2}, \hat{I}_2) \right| + \sum_{S^{e_2}} \sum_{S^{e_1}} \left| \frac{1}{|S^{e_2}|} P(s_{t+1}^{e_1}|s_t^{e_1}, a_t) - \frac{1}{|S^{e_1}|} P(s_{t+1}^{e_2}|s_t^{e_2}, a_t) \right| \right] \\
+ R_{max} + \gamma \max_{e_1,e_2,t} |V^{e_1}(s_{t+1}^{e_1}|\hat{I}_1) - V^{e_2}(s_{t+1}^{e_2}|\hat{I}_2)|
\end{aligned}
\tag{18}
$$

As Equation 18 always holds for any $s_t^{e_1}, s_t^{e_2}$, there is:

$$
\begin{aligned}
|V^{e_1}(s_t^{e_1}|\hat{I}_1) - V^{e_2}(s_t^{e_2}|\hat{I}_2)| \le \max_{e_1,e_2,t} |V^{e_1}(s_t^{e_1}|\hat{I}_1) - V^{e_2}(s_t^{e_2}|\hat{I}_2)| \le \frac{R_{max}}{(1-\gamma)^2} \cdot \\
\left[ \sum_{a_t} \left| \pi^{e_1}(a_t|s_t^{e_1}, \hat{I}_1) - \pi^{e_2}(a_t|s_t^{e_2}, \hat{I}_2) \right| + \sum_{S^{e_2}} \sum_{S^{e_1}} \left| \frac{1}{|S^{e_2}|} P(s_{t+1}^{e_1}|s_t^{e_1}, a_t) - \frac{1}{|S^{e_1}|} P(s_{t+1}^{e_2}|s_t^{e_2}, a_t) \right| \right] \\
+ \frac{R_{max}}{1-\gamma}
\end{aligned}
\tag{19}
$$

With Assumption 3.2, there is:

$$
\begin{aligned}
\max_{e_1,e_2} |V^{e_1}(s_t^{e_1}|\hat{I}_1) - V^{e_2}(s_t^{e_2}|\hat{I}_2)| \le \\
\frac{R_{max}}{(1-\gamma)^2} \cdot \left[ L_\psi |A| \left\| \hat{I}_1 - \hat{I}_2 \right\| + \max_{e_1,e_2} |S|^2 \left| \frac{P(s_{t+1}^{e_1}|s_t^{e_1}, a_t)}{|S^{e_2}|} - \frac{P(s_{t+1}^{e_2}|s_t^{e_2}, a_t)}{|S^{e_1}|} \right| \right] + \frac{R_{max}}{1-\gamma}
\end{aligned}
\tag{20}
$$

$\square$

## A.2 PROOF OF PROPOSITION 3.4

*Proof.* With Definition 3.1, we can find an interesting property of learning in such MDPs of multiple environments. That is, with the correct reward setting, maximizing the return is equal to finding the common point of training tasks. It is shown as follows.

Consider the optimizing objective function of reinforcement learning, i.e.,

$$
\max_{\pi(\hat{I})} \mathbb{E}_{e \in \mathcal{E}} \left[ \sum_{t \ge 0} R^e(s_t^e, a_t|I) \right]
\tag{21}
$$

We consider the agent is trained with sparse reward as follows:

$$
R(s_t^e, a_t|I) = \begin{cases} 1, & \text{if } I \in s_t^e \\ 0, & \text{otherwise} \end{cases}
\tag{22}
$$

Here $I \in s_t^e$ means that $s_t^e$ is the final goal state of the task and the component $\psi_t(I)$ of representation $I$ in this environment can be observed in $s_t^e$. For a common example, if a task is to pick up a red ball, only when the agent observes the red ball and correctly interacts with it, will it obtain a reward.

With Equation 22, there is:

$$
\begin{aligned}
&\mathbb{E}_{e \in \mathcal{E}, \tau^e \sim \pi^e}\big[\sum_{t \geq 0} R^e(s_t^e, a_t | I)\big] \\
=&\frac{1}{|\mathcal{E}|} \sum_{e \in \mathcal{E}} \sum_{\tau^e} \pi^e(\tau^e | \hat{I}) \sum_{t \geq 0} \gamma^t R(s_t^e, a_t | I) \\
=&\frac{1}{|\mathcal{E}|} \sum_{e \in \mathcal{E}} \sum_{\tau^e : I \in s_T^e} \gamma^T \pi^e(\tau^e : I \in s_T^e | \hat{I})
\end{aligned}
\tag{23}
$$

where $T$ is the terminate time and $s_T^e$ is the final state.

In reinforcement learning, the policy $\pi^e(\tau^e : I \in s_T^e | \hat{I})$ can be seen as the probability of given $\hat{I}$, $e$, and $\tau^e$ then $I$ occurs. Thus $\pi^e(\tau^e : I \in s_T^e | \hat{I})$ can be seen as $P^\pi(I, \tau^e | e, \hat{I})$. Then there is:

$$
\begin{aligned}
&\frac{1}{|\mathcal{E}|} \sum_{e \in \mathcal{E}} \sum_{\tau^e : I \in s_T^e} \gamma^T \pi^e(\tau^e : I \in s_T^e | \hat{I}) \\
=&\frac{1}{|\mathcal{E}|} \sum_{e \in \mathcal{E}} \sum_{\tau^e : I \in s_T^e} \gamma^T P^\pi(I, \tau^e | e, \hat{I}) \\
\leq&\frac{1}{|\mathcal{E}|} \sum_{e \in \mathcal{E}} \sum_{\tau^e} P^\pi(I, \tau^e | e, \hat{I}) \\
=&\frac{1}{|\mathcal{E}|} \sum_{e \in \mathcal{E}} P^\pi(I | e, \hat{I}) \\
=&\sum_{e \in \mathcal{E}} P^\pi(I | e, \hat{I}) P(e) \\
=&P^\pi(I | \hat{I})
\end{aligned}
\tag{24}
$$

Then maximizing $\mathbb{E}_{e \in \mathcal{E}, \tau^e \sim \pi^e}[\sum_{t \geq 0} R^e(s_t^e, a_t | I)]$ will also maximizing $P^\pi(I | \hat{I})$. They are equal objective functions when optimizing. $\qquad\square$

It means that, ideally, if the agent can learn successfully in the same task in different environments, training in similar tasks in multiple environments will be motivated to find the final states that have the shared task representation. This process in RL can be seen as extracting shared representation $I$ by aligning the given one $\hat{I}$ and ignoring the change of the backgrounds of environments. After correctly learning the shared invariant representation, the agent can correctly complete the identified tasks by the goal-conditioned policy with correct $\hat{I}$.

However, this ideal situation holds when the agent can learn successfully and all the environments can be obtained when training. In generalization tasks, especially in real-world complex tasks, the environments and tasks are usually inaccessible. Thus, in this paper, we will focus on how to deal with the problem in finite training environments and generalize to unseen new ones.

### A.3 PROOF OF THEOREM 4.1

*Proof.* We will begin with the error between the original objective expectation and the objective after randomization. Here we consider a more complex case in the distributions of the original environments and the randomized environments are different, which will be more practical.

$$
\left| \mathbb{E}_{e \in \mathcal{E}, \tau^e \sim \pi^e} [\sum_{t \geq 0} R^e(s_t^e, a_t | I)] - \mathbb{E}_{\hat{\xi}_t \in \Xi, \hat{\tau} \sim \hat{\pi}} [\sum_{t \geq 0} \gamma^t R(\hat{s}_t, a_t | I)] \right|
$$

$$
= \left| \sum_{e \in \mathcal{E}} \rho(e) \sum_{\tau^e} \pi^e(\tau^e | \hat{I}) \sum_{t \geq 0} \gamma^t R(s_t^e, a_t | I) - \sum_{\hat{\xi} \in \Xi} \rho(\hat{\xi}) \sum_{\hat{\tau}} \hat{\pi}(\hat{\tau} | \hat{I}) \sum_{t \geq 0} \gamma^t R(\hat{s}_t, a_t | I) \right| \tag{25}
$$

where $\rho(\cdot)$ is the distribution function.

Denoting $\sum_{\tau^e} \pi^e(\tau^e | \hat{I}) \sum_{t \geq 0} \gamma^t R(s_t^e, a_t | I)$ as $J(e)$ and $\sum_{\hat{\tau}} \hat{\pi}(\hat{\tau} | \hat{I}) \sum_{t \geq 0} \gamma^t R(\hat{s}_t, a_t | I)$ as $J(\hat{\xi})$, Equation 25 can be rewritten as inner product form as follows:

$$
(25) = \left| \left\langle \vec{\rho}_e, \vec{J}_e \right\rangle - \left\langle \vec{\rho}_{\hat{\xi}}, \vec{J}_{\hat{\xi}} \right\rangle \right| \tag{26}
$$

where $\vec{\rho}_e, \vec{J}_e, \vec{\rho}_{\hat{\xi}}, \vec{J}_{\hat{\xi}}$ are all vectors of $|\mathcal{E}|$-dimension. Subsequently, similar with Equation 9 there is:

$$
\begin{aligned}
&\left| \left\langle \vec{\rho}_e, \vec{J}_e \right\rangle - \left\langle \vec{\rho}_{\hat{\xi}}, \vec{J}_{\hat{\xi}} \right\rangle \right| \\
&= \left| \left\langle \vec{\rho}_e, \vec{J}_e \right\rangle - \left\langle \vec{\rho}_e, \vec{J}_{\hat{\xi}} \right\rangle + \left\langle \vec{\rho}_e, \vec{J}_{\hat{\xi}} \right\rangle - \left\langle \vec{\rho}_{\hat{\xi}}, \vec{J}_{\hat{\xi}} \right\rangle \right| \\
&\leq \left| \left\langle \vec{\rho}_e, \vec{J}_e \right\rangle - \left\langle \vec{\rho}_e, \vec{J}_{\hat{\xi}} \right\rangle \right| + \left| \left\langle \vec{\rho}_e, \vec{J}_{\hat{\xi}} \right\rangle - \left\langle \vec{\rho}_{\hat{\xi}}, \vec{J}_{\hat{\xi}} \right\rangle \right| \\
&= \underbrace{\left| \left\langle \vec{\rho}_e, \vec{J}_e - \vec{J}_{\hat{\xi}} \right\rangle \right|}_{①} + \underbrace{\left| \left\langle \vec{\rho}_e - \vec{\rho}_{\hat{\xi}}, \vec{J}_{\hat{\xi}} \right\rangle \right|}_{②}
\end{aligned} \tag{27}
$$

With Hold's inequality, there is:

$$
② \leq \left\| \vec{\rho}_e - \vec{\rho}_{\hat{\xi}} \right\|_1 \cdot \left\| \vec{J}_{\hat{\xi}} \right\|_\infty \tag{28}
$$

As $\sum_{\hat{\tau}} \hat{\pi}(\hat{\tau} | \hat{I}) \sum_{t \geq 0} \gamma^t R(\hat{s}_t, a_t | I) \leq \frac{\hat{R}_{max}}{1-\gamma}$, where $\hat{R}_{max} = \max_{e, \hat{\xi}_t, a_t, I} \{ R(s_t^e, a_t | I), R(\hat{s}_t, a_t | I) \}$ with Pinsker's inequality, there is:

$$
\begin{aligned}
② &\leq \left\| \vec{\rho}_e - \vec{\rho}_{\hat{\xi}} \right\|_1 \cdot \left\| \vec{J}_{\hat{\xi}} \right\|_\infty \\
&\leq \frac{\hat{R}_{max}}{1-\gamma} D_{TV}(\rho(e), \rho(\hat{\xi})) \\
&\leq \frac{\hat{R}_{max}}{1-\gamma} \sqrt{2 D_{KL}(\rho(e) || \rho(\hat{\xi}))}
\end{aligned} \tag{29}
$$

Similarly, for ① there is:

$$
\begin{aligned}
① &\leq \| \vec{\rho}_e \|_1 \cdot \left\| \vec{J}_e - \vec{J}_{\hat{\xi}} \right\|_\infty \\
&\leq \max_{e, \hat{\xi}} |J(e) - J(\hat{\xi})|
\end{aligned} \tag{30}
$$

By our non-parameterized randomization setting, $|J(e) - J(\hat{\xi})|$ can be aligned, hence can calculated inner the summation, i.e.,

$$\max_{e,\hat{\xi}} |J(e) - J(\hat{\xi})| \leq \frac{\delta_{max}}{1 - \gamma} \tag{31}$$

where $\delta_{max} = \max_{e,\hat{\xi}} |R(s_t^e, a_t|I) - R(\hat{s}_t, a_t|I)|$

Then there is:

$$\left| \mathbb{E}_{e \in \mathcal{E}, \tau^e \sim \pi^e} [\sum_{t \geq 0} R^e(s_t^e, a_t|I)] - \mathbb{E}_{\hat{\xi}_t \in \Xi, \hat{\tau} \sim \hat{\pi}} [\sum_{t \geq 0} \gamma^t R(\hat{s}_t, a_t|I)] \right|$$
$$\leq \frac{1}{1 - \gamma} \left( \hat{R}_{max} \sqrt{2 D_{KL}(\rho(e)||\rho(\hat{\xi}))} + \delta_{max} \right) \tag{32}$$

i.e.,

$$\mathbb{E}_{e \in \mathcal{E}, \tau^e \sim \pi^e} [\sum_{t \geq 0} R^e(s_t^e, a_t|I)] \geq$$
$$\mathbb{E}_{\hat{\xi}_t \in \Xi, \hat{\tau} \sim \hat{\pi}} [\sum_{t \geq 0} \gamma^t R(\hat{s}_t, a_t|I)] - \frac{1}{1 - \gamma} \left( \hat{R}_{max} \sqrt{2 D_{KL}(\rho(e)||\rho(\hat{\xi}))} + \delta_{max} \right) \tag{33}$$

$\square$

Notice that Equation 31 does not hold in this form if the randomization is executed by randomizing a parameterized environment model. If the environment model cannot align with the generalizing environments, it should introduce an additional virtual term like Equation 27, meaning that the error bound will be an additional term. It indicates that utilizing a parameterized model to make randomization may cause additional risks in environmental generalization tasks, explaining why DR-like methods perform poorly in such tasks.

## B    DETAILS OF EXPERIMENT SETTING

### B.1    TRAINING TASKS FOR STABLE LEARNING

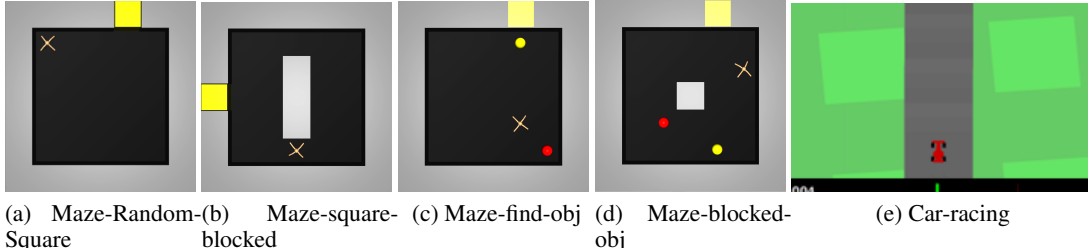

(a) Maze-Random-Square    (b) Maze-square-blocked    (c) Maze-find-obj    (d) Maze-blocked-obj    (e) Car-racing

Figure 4: Training Tasks in MuJoCo and Gym.

In MuJoCo tasks, the agent moves discretely by the oracle movement by the simulator. Every step will move the scaling of size 3 in the game. The action space is 4-dimension with 'up,left,right,down'. The size of the room is 5×5

**Maze-Random-Square**. It is a square maze. The agent and the goal door will initiate with randomized relative distance. The door has a random initial position according to the structure of the whole wall. The agent has a random initial position in the whole room.

**Maze-square-blocked**. It is a square maze with a blocked wall. The agent and the goal door will initiate with randomized relative distance. The door has a random initial position on the left and right walls. The agent has a random initial position in the whole room.

**Maze-find-obj**. It is a square maze. The agent should go to the correct object according to the given instructions. There is a door and an unrelated object as a disturbance. The relative position and viewpoint are randomized. The goal object will only occur near the wall.

**Maze-blocked-obj**. Same with 'Maze-find-obj' with a blocked wall in the middle.

**Random-Track**. CarRacing game with a randomized structure of the track. The color of the grass (background) and the zoom of the viewpoint are randomized.

**Inverse-Track**. CarRacing game with a randomized structure of the track. But it is different from the 'Random-Track'. This track will generate more irregular roads with the environment generator. The color of the grass (background) and the zoom of the viewpoint are randomized.

**Circle-No-Rand**. CarRacing game with big circle track. None of the components is randomized. It is used to show that the baselines can learn well in tasks without randomization but fail in generalization tasks.

## B.2 GENERALIZATION TASKS

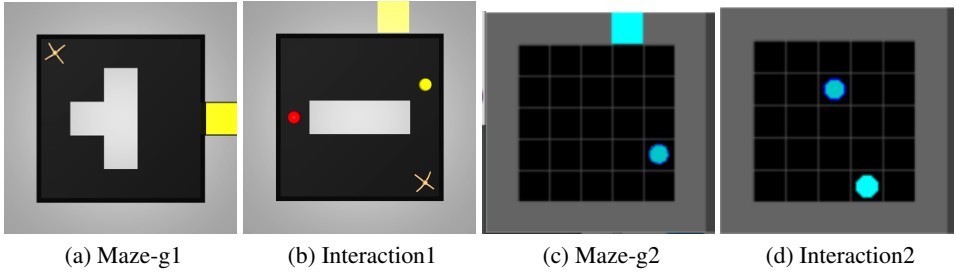

(a) Maze-g1          (b) Interaction1          (c) Maze-g2          (d) Interaction2

Figure 5: Generalization Tasks in MuJoCo and BabyAI.

**Maze-g1**. An unseen blocked maze. The initial positions are fixed. The structure of this generalization maze is different from all the seen mazes. The agent should overcome the unseen blocks and reach the door.

**Interaction1**. An unseen blocked maze when training to interact with objects. The agent should overcome the unseen blocks and find the correct object.

**Maze-g2**. An OOD environmental generalization task in the BabyAI platform. The form of the agent and the dynamics are all different. The relative positions are randomly initiated. The agent should overcome not only the difference of observation and dynamics, and the randomness to achieve the goal door.

**Interaction1**. An OOD environmental generalization task in the BabyAI platform. The form of the agent and the dynamics are all different. The distribution of the goal object is also different from the training tasks.

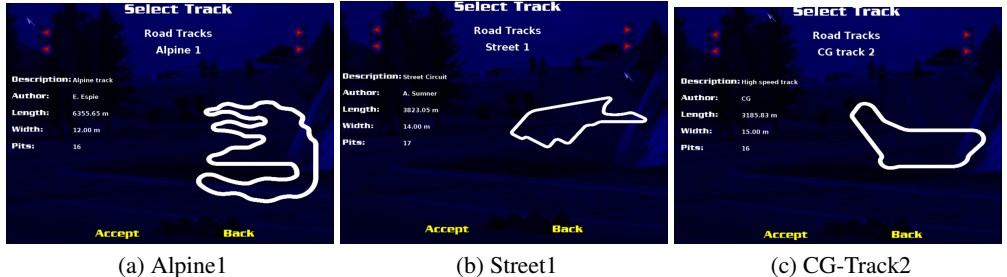

(a) Alpine1          (b) Street1          (c) CG-Track2

Figure 6: OOD Environmental Generalization Racing Tasks in Torcs.

**Tracks In Torcs**. This racing game is quite different from the training environment of the Gym, including the dynamics, the structure of the track, the background, and the viewpoint. We utilize the overhead perspective to obtain pixel observation as input. It is extremely hard to control the car by pixel input, where the action space is not perfectly aligned with the training tasks. Thus, neither the baselines nor our method can complete the racing in zero-shot. Here we utilize the racing distance as a reward and ignore the loss caused by the collision. These zero-shot agents can slowly drive along the fence of the road. The more adaptable the policy, the further it drives. The non-adaptable policy will get stuck on the side of the road and cannot drive too far.

## C  ALGORITHM

### C.1  ALGORITHM FOR DISCRETE TASKS

---
**Algorithm 1** Algorithm for Discrete Tasks

---
1: Initialize main actor parameters $\theta_a$
2: Initialize main value parameters $\theta_v$
3: Initialize multi-process actor parameters $\theta_a^i$ for $i \in [1, n]$
4: Initialize multi-process value parameters $\theta_v^i$ for $i \in [1, n]$
5: **for** episodes in 1,M **do**
6:     Reset gradients: $d\theta_a$ and $d\theta_v$
7:     **for** $i \in [1, n]$ **do**
8:         Synchronize thread-specific parameters $\theta_a^i = \theta_a$
9:         Synchronize thread-specific parameters $\theta_v^i = \theta_v$
10:        **repeat**
11:           Perform $a_t$ according to policy $\pi^i(a_t|s_t^i)$
12:           Receive reward $r_t^i$ and new state $s_{t+k}^i$
13:           $t \leftarrow t + k$
14:        **until** terminal $s_T^i$ or $t == t_{max}$
15:        Set $R = r_t^i$
16:        **for** $j \in \{t - k, t - 2k, \ldots, 0\}$ **do**
17:           $R \leftarrow r_j^i + \gamma R$
18:           Accumulate gradients w.r.t. $\theta_a$

$$d\theta_a \leftarrow d\theta_a + \frac{1}{n}\frac{k}{t}\nabla_{\theta_a} \log \pi^i(a_t|s_j^i; \theta_a)(R - V(s_j^i; \theta_v))$$

19:           Accumulate gradients w.r.t. $\theta_v$

$$d\theta_v \leftarrow d\theta_v + \frac{1}{n}\frac{k}{t}\frac{\partial}{\partial \theta_v}(R - V(s_j^i; \theta_v))^2$$

20:        **end for**
21:     **end for**
22:     Update parameters $\theta_v$ and $\theta_a$
23: **end for**

---

## C.2 Algorithm for Continuous Tasks

---

**Algorithm 2** Parallel PPO Algorithm for Continuous Tasks

---

1: Initialize main parameters $\theta$
2: Initialize multi-process parameters $\theta^i$ for $i \in [1, n]$
3: **for** episodes in 1,M **do**
4:     Reset gradients: $d\theta_a$ and $d\theta_v$
5:     **for** Parallel processes $i \in [1, n]$ **do**
6:         Synchronize thread-specific parameters $\theta^i = \theta$
7:         **repeat**
8:           Perform $a_t$ according to policy $\pi^i(a_t|s_t^i)$
9:           Receive reward $r_t^i$ and new state $s_{t+k}^i$
10:           $t \leftarrow t + k$
11:         **until** terminal $s_T^i$ or $t == t_{max}$
12:         Set $R = r_t^i$
13:         **for** $j \in \{t - k, t - 2k, \ldots, 0\}$ **do**
14:           $R \leftarrow r_j^i + \gamma R$
15:           Accumulate gradients by loss of PPO $J_{PPO}$

$$d\theta \leftarrow d\theta + \frac{1}{n} J_{PPO}(\theta^i)$$

16:         **end for**
17:     **end for**
18:     Update parameters $\theta$
19: **end for**

---

## D Network and Hype-parameters

The network is mainly a CNN with an existing actor-critic structure. The details can be seen in the code.

The main hype-parameters and details of tasks are as follows:

Table 4: Steps of every episode of different tasks

| Hyper-parameters | Value and Details |
| --- | --- |
| Step of discrete tasks | 30 |
| Step of continuous tasks | 300 |
| Action space of discrete tasks | $\uparrow, \downarrow, \leftarrow, \rightarrow$ |
| Action space of continuous tasks | 3-dimension, 1st of [-1,1] for steering, 2nd of [0,1] for gas, 3rd of [0,1] for break |
| Room size | 7*7, grids of MuJoCo and BabyAI |
| Grid size | 3*3, size in coordinates system of MuJoCo |
| Reward of discrete tasks | Sparse '1' for achieving the goal |
| Reward of continuous tasks | -0.1 for staying and 1000*racing distance/checkpoints |
| Generalizing test episodes | 500 for discrete tasks, 5 for continuous tasks |

