# OpenReview forum: "Non-Parameterized Randomization for Environmental Generalization in Deep Reinforcement Learning"
_ICLR.cc/2024/Conference — Submitted to ICLR 2024_

### Official Review · Reviewer_Et3C · 2023-10-28

**Soundness:** 1 poor
**Presentation:** 1 poor
**Contribution:** 1 poor
**Rating:** 1
**Confidence:** 4

**Summary:**

The paper considers generalisation in Reinforcement Learining such that the testing environments are "intrinsically different", though the exact definition of this is missing. The authors make an attempt of theoretical formalisation of the generalisation problem, and tackle generalisation with randomisation showing superiority of their approach in both, asymptotic performance on a distribution of the training environments, and in zero-shot generalisation to unseen environments.

**Strengths:**

Formalisation of generalisation in Reinforcement Learning is a monumental challenge. If better explained, splitting the generalisation error into optimizable and non-optimizable parts could provide an interesting point of view on the problem.

**Weaknesses:**

* The paper is extremely challenging to read with many terms not being properly defined and phrasings being too vague to grasp. I will provide more details in the 'Questions' section, but on a high level, it's actually hard to understand what the author's method proposes. The authors have some context in mind which a reader (at least me) does not have, and even after several passes, I'm not sure what some of the paragraph are about.
* The paper oversells too much. As I mentioned, formalising generalisation in RL is hard, and I appreciate the authors undertook this challenge. The authors motivate their work with practical applications, posing it as an extremely general 'environmental generalisation' problem. However, the further we go into the paper, the more assumptions are being made: action space should be shared, MDPs are goal-conditioned, (2) poses some Lipschitz-like assumptions on the policies. It is fine to introduce the assumptions and reduce the scope of the work, but I don't think the Introduction is preparing us for that.
* The paper misses important bits of related work.
  *Firstly, I don't think the paper treats related work fairly:
    * "To the best of our knowledge, our work is the first to introduce a structured framework that uniformly describe the environmental generalization problem."
    * "As there are no works that have solved environmental generalization tasks..."
" ... we propose a novel framework that tries to describe and solve the generalization RL tasks that have intrinsic environental change. To the best of our knowledge, we are the first to discuss and attempt to deal with this problem"
  * I don't think all the above is true, given the paper definition of environmental generalisation: "RL agents frequently need to adapt to diverse environmental conditions, necessitating policy adaptations to changes in state space, action space, and transition functions. This requirement, termed ”environmental generalization” in RL".
  * "A Survey of Zero-shot Generalisation in Deep Reinforcement Learning" by Kirk et al not only provides a framework for generalisation in RL, but provides a great overview of the topic.
  * From what I understood, the paper reinvented ProcGen (see "Leveraging Procedural Generation to Benchmark Reinforcement Learning" by Cobbe), which is not mentioned in the paper at all.
  * Other useful references from older research to develop authors' ideas:
    * "Autonomous shaping: knowledge transfer in reinforcement learning" by Konidaris and Barto. The paper proposes splitting the representation into two parts: agent space and problem space. The first never changes when doing transfer, the other does. This might be relevant to paper's attempt of understanding invariants when generalising.
    * "An Object-oriented representation for efficient Reinforcement Learning" by Diuk, Cohen and Littman. I brought up this paper as the authors reason about semantics of the tasks and varying objects in the environments that are not necessarily useful for completing the task's goal. "Generalizing Plans to New Environments in Relational MDPs" by Guestrin et al is in the similar vein.
    * "Transient non-stationarity and generalisation in deep reinforcement learning" by Igl et al might be interesting from the perspective of catastrophic forgetting and neurons saturation when generalising.

**Questions:**

* Could you, please, narrow down the definition of the 'environmental generalisation' in the paper. The second paragraph in the intro gives a quite general definition, but I have a feeling that the rest of the paper means something else by it. You mention 'intrinsically different' several times, could you, also, provide a definition?
* Could you define 'task-agnostic components'? Could you, please, give an example?
* What do you mean by the following? "Different from previous methods, our work intrinsically randomizes the enviroonment to build task-level augmentations and does not require a specific parameterized model". Do I understand it correctly, that for your method, you need to have access to the objects in the environmental scene? In your opinion, why is this different from varying the parameter vector of the environment (e.g. friction coefficient in MuJoCo).
* Your MDP definition implies that MDP is induced by I, but none of the MDP tuple components depend on it. Could you give more intuition about this parameter? What is this exactly? Could you give an example? Later you mention that "I stays invariable and represents the common points of the same task in different environments." Is this an additional assumption on your method/framework that tasks should have some 'common points'? What are these common points? What is the distinction between an MDP, a task and an environment in your context? Related to that, could you provide more intuition on equation (1)? Could you give an example of the reversible function in addition to the 'task-agnostic background'? What is a background of task?
* In your definition of an MDP, discount factors are shared. Do you think this is a realistic assumption? If you are motivated by the practical applications as stated in the intro, could you elaborate on the consequences of this assumption?
* Assumption 3.2 mentiones 'well-learned policies', are those optimal policies? Have you checked if the assumption in 3.2 holds in your empirical experiments?
* At the bottom of page 6, you give an example of a robot that should find an apple. You say that you can randomize all the objects non-related to the task (fridge, microwave, room structure), add some unrelated objects etc. Is this another assumption of your method? Consider, all these objects' attributes are in a vector that we can randomise, how is this different from domain randomisation? Can we call this a nonparameterised randomisation? Moreover, you mention that you might want to modify the position of the apple, why is this task-agnostic if the task is to find an apple?
* Could you explain the following sentence?  "Because compared with fixed environments, the unacceptable large variance in the dynamic learning process will disturb the gradient convergence direction". What variance do you have in mind here?

---

### Official Review · Reviewer_jHMr · 2023-10-30

**Soundness:** 2 fair
**Presentation:** 3 good
**Contribution:** 2 fair
**Rating:** 3
**Confidence:** 4

**Summary:**

This paper constructs a novel structured framework to describe environmental generalization, and analyzes the sources of environmental generalization errors. Then it proposes a non-parameterized randomization method (NPR) and theoretically proves that the method has a good generalization ability. Finally, it designs several tasks to test the generalization ability of the algorithm and demonstrates the superiority of the proposed algorithm relative to the baseline algorithms through experimental results.

**Strengths:**

The framework proposed in this paper for describing the environmental generalization problem is novel. The modeling and theoretical analysis for the generalization problem are reasonable. The non-parameterized randomization method (NPR) can solve the difficult zero-shot generalization task. The experimental tasks designed in this paper are reasonable for testing zero-sample generalization.

**Weaknesses:**

1) Compared to data augmentation and domain randomization methods, the method proposed in this paper requires stronger preconditions: 1) NPR requires the algorithm to be able to change the environment during training, and 2) the expert prior is needed to label which components of the environment are task-relevant and which are not. Both of these conditions are difficult to fulfill in practice, and thus the method has significant limitations.
2) The method is designed to utilize the two preconditions mentioned above, whereas the other algorithms in the experiment were designed without them. Therefore, I think that the generalization capability experiment in the paper is meaningless. It is unfair to the other algorithms to conduct comparison experiments under different experimental conditions.
3) “In complex generalization tasks, the common part is embedded in the environment and is not always observable.” So is it the similarity parts or the changing background of the model definition? The structured model established in the paper is unclear.

**Questions:**

1) In subsection 3.1->"Difference with Previous Models." both I and $ \psi_t(I) $ are defined as similar parts of the state space, is there any ambiguity?
2) In Table 3, the training results of NPR in 2d-maze are not as good as No-Rand, and whether the loss of the training performance is indicative of flaws in the design of the algorithm?
3) In line 6 of the subsection 5.3 Ablation Study, what does “in the same environments” refer to?
4) The data in Figure 3 are too sparse, which is different from a normal dense continuous training curve. Please provide more details of the experiment. Meanwhile, there may be instability in the results of experiments using only 3 seeds. To ensure reliability, at least 5 seeds should be used.
5) There are some errors in the formatting of the references cited in section Introduction.

---

> ### Author Response · Authors · 2023-11-14
> **Response to review jHMr**
>
> To review jHMr,
>
>
> Thank you for your review.
>
>
> + We acknowledge that we need some assumptions. But we consider it is reasonable and inevitable. If we focus on the tasks in various environments in practice, how can we build policy efficiently? In fact, none of the existing methods can adapt to the huge change in real-world scenes. What is worse, training in all the real-world tasks is unacceptable due to the expensive sampling cost. So how do you face these tasks? Here we give our idea aiming to train sufficiently in just one environment and generalize to other environments that have different physical dynamics, once for all. It will significantly save samples compared with sampling in all the environments. Consider that, if an agent can generalize to any other unseen environments without retraining, meaning another way of sample efficiency. To achieve this is extremely hard. There are no methods that can completely achieve this, neither can ours. So we do not claim to completely solve the problems, but give a novel attempt and outperform all the existing methods.
>
>
>
> + It is the generalization ability of our method important, although it needs some assumption. Because generalizing to an unseen environment is nearly impossible without additional training costs. When facing many complex RL tasks, sampling and learning all the tasks in different environments will be unacceptable and expensive. As a result, our method provides a trade-off between learning and generalization. That is, training sufficiently in one environment, then generalizing to the other environments. It is more sample efficient than learning in all these tasks and environments. Meanwhile, please pay attention that our method is in zero-shot fashion. That means when the number of testing tasks grows, our requirement for training does not increase. So we think it is more important to focus on the performance and potential of our method to achieve complex generalization tasks.
>
>
>
>
> + As for the comparison of baselines, in fact, requiring a parameterized model environment costs not less than our method. On the contrary, these methods need a more specific and elaborate physical model to provide parameters that can be randomized. For instance, in a real-world application like in a kitchen, randomizing the objects in the room (our idea) will be easier than randomizing the friction coefficient and background (the idea of baselines). Thus, compared with existing methods, our method has a superiority in generalization tasks.
>
>
>
>
> + For Q1, I represent the similar parts, \psi is the function that is related to the environment. It means that, the same I will perform differently in different environments. For instance, a red ball in BabyAI will be a little different from a red balls in MuJoCo. This setting is utilized to cover more real-world scenes.
>
> + For Q2, the No-Rand task means the training task is simple and fixed without randomness. Learning well in such tasks is very easy. Training in tasks with random is more difficult than training in a fixed task.
>
>
> + For Q3, ‘the same environment’ means the testing in a different new task comes from the same environment. Such as training in a task of MuJoCo and testing in a new task of MuJoCo. It can be seen in Table 1 ‘ Trained Envs Unseen tasks’.
>
> + For Q4, the sparse results come from taking an average of every 100 episodes. We just want to show the superiority of our method clearly. If necessary, we can train more data as you suggested and add it to the revised version of our paper.
>
> +For Q5, thank you for your suggestion, we will revise the paper.

---

> > ### Comment · Reviewer_jHMr · 2023-11-21
> >
> > 1) Regarding the third reply, the authors mentioned that the cost of the parameterized model environment method is not lower than NPR. However, NPR needs to modify the environment like the parameterized environment class method, in addition to needing the expert prior to decide whether the object is relevant to the task. So, I think NPR is more costly.
> > 2) The baselines, such as DRAC and PPO, do not require access to the environment at all. These comparisons are unfair.
> > 3)  The authors should describe the related work (e.g., domain randomization, parametric environmental model) in more detail.
> > 4) I'm also concerned about the questions raised by reviewer Et3C, and I'd like to see the authors' answers to those questions.

---

> ### Author Response · Authors · 2023-11-21
> **To review jHMr**
>
> To review jHMr,
>
> Thank you for your reply.
> 1. Firstly, the cost of NPR is not in vain. It brings strong generalization ability in different environments. The success of a large model shows the fact that the cost of training and data is necessary. Because it is impossible to achieve generalization ability without effective training. There is no free lunch. Secondly, previous methods need the parameterized physical model of an environment. The cost has been spent in manually modeling the simulator of the environment, i.e., the parameterized physical model of the environment requires both the expert prior and a precise mathematic model.  Thus previous methods can augment the data in these environments by randomizing the parameters. It is unfair to ignore the expensive cost of manually building the parameterized model and be strict with our method. For instance, the most commonly used method in DR is to randomize the dynamic coefficients and visual background. These require many prior of the environment.
>
> 2. The baselines are all trained in an environment with NPR, or they will not obtain any rewards in a new environment with intrinsic physical change. Because traditional RL settings are all integration of training and testing.  We give all the experiments in the same setting for a fair comparison. But as we said, they can hardly adapt to the environmental change.
>
> 3. We have already compared many related works not only in the Introduction but also section 'Randomization as Augmentation' related work and section 4.2 'Implementation of NPR method' with instance. With three parts of comparison from principle to method then implementation. How can we give more details?
>
> 4. We do not reply to reviewer Et3C because this reviewer does not give us any respect. Firstly, Et3C suspects our claim but gives some works that are only related to the title. If you search the paper and have a look you will find it is not related to environmental generalization anymore. For instance, the works Et3C mentioned 'An Object-Oriented Representation for Efficient Reinforcement Learning, ICML2008' and 'Autonomous Shaping: Knowledge Transfer in Reinforcement Learning, ICML2006' utilizes manual design to generalize in tabular task are not comparative works with ours. We do not consider randomly searching any paper with 'generalization' and 'transfer' in the title can be used as an argument to rebuttal our paper. Secondly, many settings like 'shared' action space are common settings in previous works. Our work breaks the assumption that all the environments share the same state space with the same intrinsic generator. It makes a further step compared with previous works.  If someone has read several papers in the RL domain, even those not related to generalization tasks will know about this fact. Thirdly, the reviewer Et3C acknowledged did not understand the paper but finally gave confidence 4. We do not think this is a fair attitude of review. The significant emotional evaluation of reviewer Et3C affects the review. We do not understand why you are concerned about the question. If you read the questions of reviewer Et3C seriously and take a look at the papers he suggested, you will find that the viewpoints of reviewer Et3C do not hold.

---

> > ### Comment · Reviewer_Et3C · 2023-11-22
> > **response**
> >
> > > We do not reply to reviewer Et3C because this reviewer does not give us any respect.
> >
> > I don't think I was disrespectful in my review.
> >
> > > Firstly, Et3C suspects our claim but gives some works that are only related to the title.
> >
> > It is quite weird to hear that. Firstly, I explicitly wrote why I suggest this or that paper. Secondly, I didn't expect the authors to compare the performance of the methods or something, but just add those to the related work section as it could help to develop an idea and it is always nice to see the research in the historical context.
> >
> > > Our work breaks the assumption that all the environments share the same state space with the same intrinsic generator. It makes a further step compared with previous works. If someone has read several papers in the RL domain, even those not related to generalization tasks will know about this fact.
> >
> > While this is common assumption, this is not always correct. There is research in RL on environments with different observation and action space.
> >
> > Given that the authors refused to engage during the rebuttal period, I keep my score unchanged.

---

> > ### Comment · Reviewer_jHMr · 2023-11-23
> >
> > Thanks for the author's responses to my comments. Based on the comments of all reviewers and the author's reply, I decided to keep my score.

---

### Official Review · Reviewer_yfA1 · 2023-10-31

**Soundness:** 3 good
**Presentation:** 3 good
**Contribution:** 2 fair
**Rating:** 3
**Confidence:** 3

**Summary:**

This paper studies the challenge of environmental generalization in RL. It introduces a formal framework that describes this challenge, identifying the main difficulty as being a non-optimizable "adaption gap" that arises from the specific dynamics and observations of the training environment. The paper proposes a non-parameterized randomization (NPR) method that augments the training environments, which it shows is equivalent to introducing an alternative objective function that offers an optimizable lower bound for the adaption gap. The authors conduct empirical evaluations on some Mujoco and CarRacing tasks.

**Strengths:**

- The paper is written clearly and well organized.
- The motivation for the problem of environmental generalization is intuitive and interesting.
- The framework provides a nice formalism for environment generalization.

**Weaknesses:**

- It seems like the NPR method boils down to randomizing task-agnostic parts of the environment to increase the diversity of the backgrounds trained on in order to improve zero-shot environment generalization. This is not a new or surprising conclusion. In fact, many recent works, such as RT-2 (Google, 2023), have shown that with increased diversity of backgrounds, zero-shot generalization can be obtained in unseen environments. Is there more nuance here that is not coming across?
- The paper makes it seem like the NPR method removes significant assumptions for training, but in practice, randomizing the non-parametrized task-agnostic parts of the environment may be even more difficult than the task-relevant parts of the environment. This is something that would not be very scalable in many real world applications.
- The comparisons in the empirical evaluation all use fewer assumptions than NPR--they do not assume access to randomizing specific aspects of the environments.

**Questions:**

See weaknesses above.

---

> ### Author Response · Authors · 2023-11-14
> **Response to review yfA1**
>
> To review yfa1,
>
> Thank you for your approval of our problem.
>
> + For the question of zero-shot generalization problems and the motivation of our work.
>
> >> Recent works show that a large model with an RL setting can achieve many complex tasks, even zero-shot generalization tasks, including RT-2, VoxPoser, Gato, and some following works. They have solved many tasks, however, have not solved all the kinds of generalization tasks. In fact, as RL aims to solve real-world tasks, the generalization tasks are quite different.
>
> >> When classified by allocation-oriented real-world tasks, from abstract aspect to specific aspect, generalization tasks can be divided into three main kinds. That is strategy generalization of abstract decisions, skill-level generalization of recombination tasks, and manipulation generalization of different low-level robots. When classified by RL modeling, generalization tasks can be divided into two categories, i.e., focusing on task change (reward function or transition change) or environmental change (physical model of state space and action space change). A survey [1] of zero-shot generalization in deep RL also gives another view of IID and OOD generalization.
>
> >> As for our work, we focus on generalization across different environments, where the intrinsic physical generators of the environments are different, which has not been solved by existing works yet. The difference between our work and existing works that focus on tasks with background change is that the environmental change in our work will impact the policy execution, but the other will not. For instance, a prevailing task [2] that previous works claimed with environment change is controlling the MuJoCo robot with the background changing in the DMControl platform. The changing backgrounds consist of task-agnostic pictures, but it does not change the structure of the environment. That means that if an agent learns an action sequence in the training tasks, it can also complete the task with the sequence. The intrinsic model of the tasks does not change, meaning that the tasks are intrinsically the same.
>
> >> On the contrary, the environmental change in our tasks requires the policy to adapt to different dynamics and physics in different environments. That is why the tasks are quite difficult. As a result, our work does not claim to completely solve these tasks, but just gives an attempt and proposes an idea. Solving all these kinds of tasks means the agent can adapt to any OOD tasks, so should also be used in real-world applications in zero-shot.
>
> >> Meanwhile, our method does not conflict with the existing large models. Our method is in a goal-conditioned setting, which can be used as the low-level executing policy of large models by receiving high-level abstract instructions.
>
> + For the question of assumption and settings in our method.
>
> >> For our NPR method, we acknowledge that we need some assumptions. But we consider it is reasonable and inevitable. When facing many complex RL tasks, sampling and learning all the tasks in different environments will be unacceptable and expensive. As a result, our method provides a trade-off between learning and generalization. That is, training sufficiently in one environment, then generalizing to the other environments. It is more sample efficient than learning in all these tasks and environments. Meanwhile, please pay attention that our method is in zero-shot fashion. That means when the number of testing tasks grows, our requirement for training does not increase. So we think the cost in the training environment is acceptable.
>
> + For the question of comparison of other baselines.
>
> >> As for the comparison of baselines, in fact, requiring a parameterized model environment costs not less than our method. On the contrary, these methods need a more specific and elaborate physical model to provide parameters that can be randomized. For instance, in a real-world application like in a kitchen, randomizing the objects in the room (idea of ours) will be easier than randomizing the friction coefficient and background colors (idea of baselines). Meanwhile, the methods that augment the data after samples can not represent the intrinsic change of environments, so they perform poorly. Thus, compared with existing methods, our method has a superiority in generalization tasks.
>
>
>
> [1] Kirk, Robert, et al. "A survey of zero-shot generalisation in deep reinforcement learning." Journal of Artificial Intelligence Research 76 (2023): 201-264.
> [2] Fan, Linxi, et al. "SECANT: Self-Expert Cloning for Zero-Shot Generalization of Visual Policies." International Conference on Machine Learning. PMLR, 2021.

---

### Meta-Review · Area_Chair_9c4x · 2023-12-03

**Metareview:**

The paper proposes a theoretical framework to capture the difficulties of reinforcement learning generalization, called "non-parameterized randomization" (NPR).

Reviewers gave in total a (3, 3, 1) score, which leads to a clear reject opinion. Reviewers unanimously agreed that the paper does not take into account previous work (RT-2, ProcGen, Survey of RL generalization, etc.). Furthermore, a common opinion shared by reviewers (yfA1, jHMr) is that a user must label the irrelevant + relevant parts of the environment, which is unscalable in practice. Reviewers also raised numerous concerns about the experimental evaluations.

Theoretically explaining generalization in reinforcement learning (RL) is a difficult task in general, and as shown by multiple previous works stating their own abstractions of the problem, there is no one-size-fits-all framework. I would recommend the authors to carefully revisit and edit their framework's assumptions.

**Justification For Why Not Higher Score:**

This was a clear reject; scores were (3, 3, 1).

**Justification For Why Not Lower Score:**

N/A

---

### Decision · Program_Chairs · 2024-01-16

Reject